# Start Task Crafting, Stay Away from Cyberloafing: The Moderating Role of Supervisor Developmental Feedback

**DOI:** 10.3390/bs14100960

**Published:** 2024-10-17

**Authors:** Man Hai, Xuyao Wu, Bingping Zhou, Ye Li

**Affiliations:** 1School of Psychology, Central China Normal University, Wuhan 430079, China; haimanpsy@163.com (M.H.); wxypp0324@gmail.com (X.W.); 2School of Education, Wenzhou University, Wenzhou 325035, China; zhoubp_psy@126.com

**Keywords:** passive cyberloafing, active cyberloafing, task crafting, supervisor developmental feedback

## Abstract

Cyberloafing as a production deviance behavior raises organizational concerns. Unfortunately, it is unknown how to minimize cyberloafing from a bottom-up perspective, particularly different types of cyberloafing. This study draws on the job crafting and dual-process theory to construct a framework for understanding the relationship between task crafting and passive–active cyberloafing, as well as their boundary condition (i.e., supervisor developmental feedback). We adopted a convenient sampling method, following a two-stage sampling with a time interval of 2 weeks. A sample of 614 full-time employed adults were recruited from the online survey. The results showed that: (1) Task crafting was negatively related to passive and active cyberloafing, respectively. (2) The impact of task crafting on passive cyberloafing rather than active cyberloafing was moderated by supervisor developmental feedback, such that task crafting had significant negative relations with passive cyberloafing when supervisor developmental feedback was higher (vs. lower). Overall, our research findings indicate that passive cyberloafing seems more sensitive to the organizational environment than active cyberloafing, thus different types of cyberloafing have different intervention strategies.

## 1. Introduction

If employees do not like their jobs, they may intentionally or unintentionally slip away from the tasks at hand and get on the Interne [1,2], that is, cyberloafing. Cyberloafing refers to employees using the Internet for non-work activities during working time [3]. Cyberloafing is commonly regarded as production deviation behavior. The company salary.com surveyed more than 3200 people on their website between February and March 2012 to find out who were the biggest time-wasters at work. The survey shows that 67 percent of employees use the company’s Internet for non-work-related purposes, ranging from one hour to ten hours a week [4]. Some employees even spend entire days cyberloafing [5]. Cyberloafing has threatened the productivity of organizations and led to significant financial losses [6]. In addition, with the development of artificial intelligence technology, work efficiency will be improved. For example, employees can use ChatGPT to help draft or edit reports and use the time saved for cyberloafing. The problem of cyberloafing may become more common in the future. Therefore, it is becoming increasingly important to find a way to minimize cyberloafing.

Early research focused on a top-down perspective on reducing cyberloafing, that is, emphasizing the organization’s strategies for managing employees. The most common strategies to reduce cyberloafing are organizational control strategies, such as regulations, monitoring, and penalties [7,8]. However, the effectiveness of these strategies is often limited and might be perceived by employees as an invasion to their privacy, and therefore are expected to have repercussions on employee behavior and loyalty [9,10]. Currently, researchers emphasize stimulating an individual’s intrinsic motivation for reducing cyberloafing, rather than relying on excessive external control [11].

Recent research has begun to call for a focus on a bottom-up perspective on how to reduce cyberloafing, which emphasizes the importance of an individual’s intrinsic motivation. Researchers have found that individual factors are important in reducing cyberloafing, such as self-control [12,13], mindfulness [14], job meaning [15], and job attachment [16]. However, there are still two aspects that have been overlooked in research on minimizing cyberloafing. First, past research has only focused on employees’ psychological factors (e.g., work meaning, job attachment) and ignored employees’ proactive behaviors. Second, recent researchers have called for attention to different types of cyberloafing behaviors as they are influenced by various factors [17]. However, past research on minimizing cyberloafing has only focused on cyberloafing as a whole, not on subcategories of cyberloafing.

Job crafting, as a proactive behavior initiated by employees, not only helps employees get rid of passive roles in the workplace but also inspires them to actively create their own job [18]. Task crafting, as a primary and important way of job crafting, involves direct and overt actions to remedy the situation when employees are frustrated and bored at work [19]. Since task crafting keeps employees engaged in their work, it is likely that it can reduce cyberloafing. In addition, the theory states that positive organizational environments, as a job resource, are conducive to job crafting. Dierdorff and Jensen (2018) found that ameliorating job crafting are associated with working in contexts rich in positive interpersonal interactions, such as supervisor support [20]. Based on this, the present study examined the role of supervisor developmental feedback in task crafting to reduce cyberloafing.

Based on the dual systems theory, cyberloafing has been categorized into passive and active cyberloafing. An employee’s unintentional and unconscious disengagement from work to engage in non-work matters is called passive cyberloafing. Conversely, an employee’s intentional and conscious disengagement from work to engage in non-work matters is called active cyberloafing. Chen et al. (2022) argued that the antecedents of these two types of cyberloafing behaviors are different [14]. Subsequently, Hai et al. (2024) found that passive rather than active cyberloafing has a negative impact on job performance [21]. This suggests that passive cyberloafing is potentially more threatening to job [22] and deserves more attention. Taken together, drawing on job crafting and dual-process theories can build a framework for understanding the relationship between task crafting and passive–active cyberloafing. Specifically, this study examined the role of task crafting and supervisor developmental feedback in minimizing passive and active cyberloafing (Figure 1).

This study contributes to the literature in three ways. First, based on the theory of task crafting [18], we provide a new bottom-up perspective for reducing cyberloafing. Second, from the person–environment interaction perspective, we explored the moderator role of supervisor developmental feedback in task crafting to reduce cyberloafing. Third, based on the dual-process theory, our study explores for the first time whether the interaction of task crafting and developmental feedback has the same effect on reducing passive and active cyberloafing [14,21]. In addition, this study provides a new cyberloafing measurement tool based on intentional and unintentional: the Active and Passive Cyberloafing Scale (APCS). In conclusion, exploring how task crafting and supervisor developmental feedback interact with passive and active cyberloafing can provide suggestions for future cyberloafing intervention practices.

## 2. Literature Review and Hypothesis Development

### 2.1. Job Crafting Theory

The need for personal control is a basic human drive. Humans respond well to having control even over seemingly small matters, and control in one’s own environment has been described as “an intrinsic necessity of life itself” [18]. Thus, one would expect that having or taking control over certain aspects of the work would be a basic human need [18]. When this need cannot be met through work, employees may intentionally or unintentionally satisfy this need through the Internet.

The job crafting theory holds that employees not only passively adapt to their jobs, but also actively create their jobs [18]. By taking control of or crafting some factors of work, even in small ways, job crafters make the job their own [18]. In this way, employees can reduce stress and meet their personal needs at work [23], thereby promoting positive job attitudes and work engagement [24] to decrease cyberloafing.

### 2.2. Based on Dual-Process Theory: Active Cyberloafing and Passive Cyberloafing

According to the dual-model theory, cyberloafing can be divided into passive cyberloafing and active cyberloafing [25]. Passive cyberloafing refers to the behaviors that employees *unintentionally* deviate from their work to non-work-related internet activities while working. For example, employees’ attention is captured by pop-up ads, news, or videos. Active cyberloafing refers to the behaviors that employees intentionally engage in non-work-related Internet activities while working, usually with obvious behavioral motivation. For example, employees actively schedule some time to take microbreaks or attend to personal affairs during work time. However, passive cyberloafing, as a common phenomenon, has not been fully considered in previous studies until recently [14,21].

The dual-process theory describes two cognitive processes: automatic processing and controlled processing [26]. The two systems work together in influencing human behaviors. The automatic processing is responsible for the emergence of an individual’s spontaneous behavior, and the controlled processing helps an individual in regulating progress towards goals and self-control. However, the two processing systems also act together, associate with each other, and impact human behavior simultaneously [26].

Cognitive processes of passive and active cyberloafing: From the perspective of the dual-process theory, whether a person can control themselves when the responses of two systems are in conflict depends on which system can “win the race”, because the automatic processing is influenced by arousal, while the controlled processing is influenced by factors such as intrinsic need and motivation [14]. When one’s automatic processing overrides the controlled processing, impulsive, unplanned, and unconscious behavior may arise, such as passive cyberloafing. Researchers hold that passive cyberloafing is not triggered by an individual’s subjective intention but is related to the imbalance of the dual systems in their brain [14]. On the contrary, when one’s controlled processing overrides the automatic processing, purposeful, planned, and conscious behavior may arise, such as active cyberloafing. Some researchers’ attempt to explain cyberloafing in terms of planned behavior theory, which supports this view [5].

How to reduce passive and active cyberloafing: Based on the dual-process theory, stress and external novel stimuli are important factors in activating automatic processing systems to induce passive cyberloafing. For example, research has found that job insecurity can induce passive cyberloafing through spontaneous mind wandering [21]. Kang and Kurtzberg (2019) found that external stimuli (such as gaming or web searches or photos) caused employees to unconsciously deviate from their work [27]. Conversely, mindfulness reduces spontaneous mind wandering and passive cyberloafing [14]. Therefore, the core of lowering passive cyberloafing is to reduce the loss of resources and increase the attractiveness of work and prevent employees from being distracted.

An individual’s intrinsic need to enhance the control processing system leads to active cyberloafing. It should be noted that the intentions for active cyberloafing can vary (such as the need for recovery and the need to balance family and work), but a common situation is that if the job cannot meet the intrinsic needs of the employee, they will choose the Internet to satisfy themselves. For example, Zacher (2014) found that employees who frequently took active breaks to replenish their energy were often those who could not replenish their energy through work [28]. Cheng (2018) found that misfit between personal abilities and job requirements decreases employee harmonious passion in work, thus leading to cyberloafing [29]. Therefore, if active cyberloafing is to be reduced, employees’ intrinsic needs need to have the opportunities to be met through tasks.

Task crafting creates an opportunity for employees to remove elements of work they do not like, add elements of work they do like, and begin to explore how their unmet intrinsic needs can be met through work [18]. It can help reduce the stress employees feel, increase job attraction, and promote meaningful and nourishing relationships between employees and their jobs [18]. Therefore, we believe that task crafting is likely to reduce passive cyberloafing and active cyberloafing, respectively, and that developmental feedback as an external resource supports the above processes.

### 2.3. The Relationship between Task Crafting and Passive–Active Cyberloafing

Task crafting describes modifications of the number, scope, or type of job duties, such as taking on new tasks, changing work processes, or devoting extra time to certain aspects of the job [18]. Our study hypothesizes that task crafting can reduce passive cyberloafing. Through task crafting, employees can remove inappropriate work processes and even tasks to reduce the obstacles and limitations they encounter in their work. These proactive changes that employees make in their job may not only change their attitude in stress, but also change the actual perceived stress. This prevents the activation of automatic systems due to the depletion of personal resources. In addition, task crafting allows employees to complete their work in their own way, increasing the attractiveness and freshness of the job, which can encourage employees to focus on their tasks and reduce unconscious distractions at work, such as passive cyberloafing.

Our study hypothesizes that task crafting reduces active cyberloafing. Task crafting helps to improve person–job fit, enhance employees’ motivation, and increase work engagement [30]. It gives employees the opportunity to fulfill themselves through work rather than non-work activities. In addition, task crafting links employees to understand their work meaning [31]. Meaning at work promotes the development of employees’ potential and self-actualization at work. When the task crafter’s potential is developed through work, he/she will further expand his/her work and find new and better ways of doing things [32]. When employees can be satisfied through their work, they are more likely to give up active cyberloafing. Therefore, we propose the following:

**H1:** *Task crafting will be negatively related to (a) passive cyberloafing and (b) active cyberloafing*.

### 2.4. The Moderator Role of Supervisor Developmental Feedback

Job crafting as a contextually embedded phenomenon suggests that the outcomes individuals experience as a result of crafting their roles are likely shaped by the context in which they work because work contexts are known to influence the functional relationships between individual-level variables [20]. Developmental feedback refers to the extent to which supervisors provide employees with helpful and useful information to facilitate learning and development on the job [32]. Supervisor developmental feedback is future-oriented and there is no pressure for a specific outcome [32]. Therefore, it can be used as an external resource to support employees’ task crafting process. Thus, we suggest that developmental feedback may moderate the relationship between task crafting and cyberloafing. Below we will specify how developmental feedback moderates the relationship between task crafting and passive and active cyberloafing, respectively.

Supervisor developmental feedback moderates the relationship between task crafting and passive cyberloafing. First, supervisor developmental feedback as a powerful situational support [20] can help employees cope with the challenges in the process of task crafting. Third-party perspectives from supervisors can help crafters view stress as a challenge, which helps employees adopt a problem-oriented approach to dealing with crises and avoid stress-activated automatic systems, thereby reducing passive cyberloafing behaviors [14]. In addition, through developmental feedback, when employees have a clear picture of their situation and resources, they are able to reduce the depletion of resources triggered by stagnating too long at obstacles, thus facilitating the progress of task crafting. Second, supervisor developmental feedback serves as a form of external feedback that helps employees identify their habitual behaviors [33]. Habitual behaviors usually take a while to change, and external reminders as staged calibrations help employees accelerate the abandonment of habitual, non-adaptive behavioral patterns and the establishment of new, adaptive behavioral patterns, which in turn help employees focus on tasks rather than Internet stimuli. Thus, we hypothesize that supervisor developmental feedback buffers the relationship between task crafting and passive cyberloafing. That is, high levels of developmental feedback are more helpful for employees to receive support and reminders from their supervisors, reducing stress levels during the crafting process and thus reducing passive cyberloafing. Conversely, task crafting has no significant effect on passive cyberloafing.

Supervisor developmental feedback moderates the relationship between task crafting and active cyberloafing. Task crafting as a creative work activity requires employees to try new tasks and approaches based on their interests and abilities [20,34]. During the crafting process, supervisor developmental feedback serves as a developmental guide to help employees further identify and clarify job demands and crafting goals, and match their job strengths and potential resources, providing opportunities to maximize the satisfaction of employees’ intrinsic needs. At the same time, as the supervisor acts as the representative of the organization, supervisor developmental feedback can also promote employees to align personal crafting goals with organizational goals [35], thus achieving the integration of personal resources and job resources. It provides assistance for employees to maximize their individual value and creativity at work [32,36]. Positive job outcomes as positive feedback will help crafters invest their time and energy in tasks rather than in cyberloafing. Based on this, we argue that developmental feedback can buffer the relationship between task crafting and active cyberloafing. That is higher levels of developmental feedback provide employees with a supportive environment that enhances the effects of task crafting and helps employees’ intrinsic needs to be further satisfied through their work, thus reducing active cyberloafing. Conversely, task crafting may have no significant effect on active cyberloafing. Therefore, we propose the following:

**H2:** *Supervisor developmental feedback moderates the negative relationship between task crafting and (a) passive cyberloafing and (b) active cyberloafing, respectively, such that the relationship is stronger when the supervisor developmental feedback is greater*.

## 3. Study 1: The Reliability and Validity of the Active and Passive Cyberloafing Scale

This study is used to illustrate the reliability and validity of the Active and Passive Cyberloafing Scale (APCS). For more information on the application of this scale to employees, please refer to previous published studies [21,37].

### 3.1. Method

#### 3.1.1. Participants and Procedure

To examine the reliability of the Active and Passive Cyberloafing Scale (APCS), we conducted an online study with 250 full-time employees. Participants who failed the attention test items were excluded. Finally, 224 valid responses were obtained, with a questionnaire recovery rate of 89.6%.

#### 3.1.2. Measures

In order to check the reliability and validity of the APCS, we conducted an online study on 250 full-time employees. Participants were assured that participation in this survey was completely voluntary, anonymous, and confidential. Respondents could withdraw at any point; however, to encourage responses, participants were rewarded with CNY 2 each time for their participation. We deleted the participants who failed one attention test item and finally obtained 224 valid responses, with an effective recovery rate of 89.6%. The mean age of the sample was 25.8 (SD = 5.72) years, 40.6% were males, the average working hours per day were 8.4 (SD = 3.21) hours, and the average tenure was 2.98 (SD = 3.20) years.

*Lim’s Cyberloafing* was measured with the cyberloafing scale developed by Lim (2002) [3]. This scale is one of the most widely used scales in the field of cyberloafing. Sample items are “browsing entertainment related websites” and “shop online for personal goods”. Participants responded on a six-point scale (1 = hardly ever, 6 = very often). The Cronbach’s alpha for this scale was 0.91.

*Cyberloafing* was measured with the Active and Passive Cyberloafing Scale [37], which consisted of passive cyberloafing (five items, e.g., “I am vulnerable to the interference of Internet stimuli and put down the task at hand”; α = 0.91) and active cyberloafing (five items, e.g., “I purposefully complete other tasks that unrelated to work through the Internet”; α = 0.90). Participants responded on a six-point scale (1 = hardly ever, 6 = very often). For more details, please see Appendix A.

*Goal clarity* was measured with the three-item subscale from the Goal Setting Questionnaire of Lee et al. (1991) [38]. Sample items are “I have specific, clear goals to aim for on my job”, “I have specific, clear goals to aim for on my job” and “If I have more than one goal to accomplish, I know which ones are most important and which are least important”. Participants responded on a five-point scale (1 = strongly disagree, 5 = strongly agree). The Cronbach’s alpha for this scale was 0.85.

*Job performance* was measured with Gong et al.’s (2009) four-item scale [39]. Sample items are “I always complete job assignments on time” and “I am one of the best employees in my work unit”. Participants responded on a five-point scale (1 = strongly disagree, 5 = strongly agree). The Cronbach’s alpha for this scale was 0.81.

### 3.2. Result

The means, standard deviations, and correlations among the variables are presented in Table 1. *Reliability*. The results of the study showed that the reliability of the total cyberloafing scale and the active and passive cyberloafing subscales were 0.93, 0.91, and 0.90, respectively. *Aggregation validity*. The results showed that the correlation between Lim’s cyberloafing and cyberloafing was 0.75. The correlation between Lim’s cyberloafing and active cyberloafing, passive cyberloafing was 0.77, and 0.62, respectively.Lim’s cyberloafing scale was defined as an active behavior in his study. In our study, the results showed that the correlation between Lim’s cyberloafing scale and the active cyberloafing subscale was higher (*r* = 0.77, *p* < 0.01) than that of the passive cyberloafing subscale (*r* = 0.62, *p* < 0.01), which also indicated that active cyberloafing subscale had good aggregation validity. *Criterion validity*. Passive cyberloafing had a significant negative correlation with goal clarity (*r* = −0.16, *p* < 0.05) and job performance (*r* = −0.20, *p* < 0.01). Active cyberloafing had no significant correlation with goal clarity (*r* = −0.04, *p* > 0.05) and job performance (*r* = −0.12, *p* > 0.05). It showed that active cyberloafing was different from passive cyberloafing.

In terms of demographic variables, cyberloafing had a significant negative correlation with rank (*r* = −0.14, *p* < 0.05). The higher the rank, the less cyberloafing, especially passive cyberloafing (*r* = −0.16, *p* < 0.05). There was no significant correlation between cyberloafing and gender, age, education level, rank, or tenure, respectively.

The results indicated that the APCS can be used to measure cyberloafing behavior in organizations.

Based on the dual systems theory, we have developed the Active and Passive Cyberloafing Scale. Therefore, in the following research, we plan to apply this scale to test the hypothesized model.

## 4. Study 2: The Moderating Effect of Developmental Feedback on the Relationship between Task Crafting and Cyberloafing

### 4.1. Method

#### 4.1.1. Participants and Procedure

The data collection involved a two-wave survey using the online platform Tencent Wenjuan (Tencent Wenjuan; Tencent Inc., CN, https://wj.qq.com). Participants were assured that participation in this survey was completely voluntary, anonymous, and confidential. The study had a two-wave design with a two-week time lag. At Time 1, 865 participants reported their job crafting and supervisor developmental feedback. At Time 2, 646 participants who responded at Time 1 were asked to report their cyberloafing behaviors and demographics. Respondents could withdraw at any point, however, to encourage response, participants were rewarded with CNY 5 each time for their participation. Our final sample consisted of 614 full-time employees. Of these, 32 employees’ data were not accepted due to incomplete questionnaire responses. The mean age of the sample was 32.74 years (SD = 9.30 years), 36.6% were males, and 89.3% were white-collar workers.

#### 4.1.2. Measures

*Cyberloafing* was measured with the Active and Passive Cyberloafing Scale [37]. This is the same as Study 1.

*Task crafting* was measured with the five-item subscale from the Job Crafting Questionnaire of Slemp and Vella-Brodrick [40]. One sample item is “Introduce new approaches to improve your work.” Participants responded on a six-point scale (1 = hardly ever, 6 = very often). The Cronbach’s alpha for this measure was 0.83.

*Supervisor developmental feedback* was measured with Zhou et al.’s (2003) three-item scale [32]. To better conform to the language habits of the Chinese employees, we changed the reverse-scored item [41]. Sample items are “My immediate supervisor never gives me developmental feedback,” and “My immediate supervisor often gives me developmental feedback”. Participants responded on a five-point scale (1 = strongly disagree, 6 = strongly agree). The Cronbach’s alpha for this measure was 0.88.

### 4.2. Results

#### 4.2.1. Confirmatory Factor Analyses

The measurement model of the study variables was tested using confirmatory factor analysis (CFA). Because of strong correlations between two dependent variables, the discriminant of four constructs was tested by contrasting a three-factor model. The four-factor model (i.e., task crafting, supervisor developmental feedback, passive cyberloafing, and active cyberloafing) was a better fit with the data (*χ*^2^ (129) = 252.63, *χ*^2^/*df* = 1.96, comparative fit index (CFI) = 0.98, Tucker–Lewis Index (TLI) = 0.98, and root mean square error of approximation (RMSEA) = 0.04) than the three-factor model in which passive and active cyberloafing loaded onto a single latent factor (*χ*^2^ (132) = 1193.14, *χ*^2^/*df* = 9.04, CFI = 0.86, TLI = 0.83, and RMSEA = 0.11). All the factor loadings were significant. This indicated that passive and active cyberloafing were different constructs, and our hypothesized four-factor model could move on to test the proposed hypotheses.

#### 4.2.2. Descriptive Statistics

The means, standard deviations, and correlations among the variables are presented in Table 2. As expected, task crafting was negatively correlated with passive cyberloafing (*r* = −0.14, *p* < 0.01) and active cyberloafing (*r* = −0.12, *p* < 0.01), respectively.

#### 4.2.3. Hypothesis Testing

We conducted SEM (structural equation modeling) to test the hypotheses. Mplus 7.4 was used for data analysis of the model [42,43]. Step 1: Check the baseline SEM model without latent interaction terms. The results showed that the model fit was good: *χ*^2^ = 1.96, RMSEA = 0.04, CFI = 0.98, TLI = 0.98. Step 2: Compare the SEM model containing the interaction terms with the baseline SEM model. In the LMs-based mediated effect analysis, the smaller the AIC (Akaike Information Criterion), the better. This is because smaller information criterion values represent smaller model complexity and higher model fit, and that BIC (Bayesian Information Criterion) values do not need to be considered [42,44]. The model index of the SEM model containing the interaction terms (AIC = 27,130.23, BIC = 27,404.26) was better than the baseline SEM model (AIC = 27,134.45, BIC = 27,399.65). The decrease in AIC indicated that the model had improved [44]. Even after adding three control variables commonly used in previous studies, such as gender, age, and rank, the AIC value was still found to decrease. The model index of the SEM model containing the interaction terms (AIC = 27,085.63, BIC = 27,386.19) was better than that of the baseline SEM model (AIC = 27,087.30, BIC = 27,379.02), which reflected the robustness of the research model. The result indicated that supervisor developmental feedback can indeed play a moderating role between task crafting and passive and active cyberloafing, respectively. Step 3: Report the formal research results. As shown in Figure 2, the main effects of task crafting on passive and active cyberloafing were significant. Specifically, task crafting was negatively related to passive cyberloafing (*β* = −0.18, *p* < 0.05, 95%CI [−0.27, −0.09]) and negatively related to active cyberloafing (*β* = −0.15, *p* < 0.05, 95%CI [−0.24, −0.05]), supporting Hypotheses 1a and 1b, respectively.

Per Figure 3, the moderating effect of supervisor developmental feedback on the relationship between task crafting and passive cyberloafing was significant (*β* = −0.11, *p* < 0.05, 95%CI [−0.19, −0.04]). Further, task crafting had a weaker negative relation with passive cyberloafing when supervisor developmental feedback was higher (simple slope = −0.29, *p* < 0.05, 95%CI [−0.42, −0.16]) than low (simple slope = −0.07, *p* > 0.05, 95%CI [−0.18, 0.05]), supporting Hypothesis 2a. However, the moderating effect of supervisor developmental feedback on the relationship between task crafting and active cyberloafing (*β* = −0.08, *p* = 0.06, 95% CI [−0.16, 0.01]) was not significant. Hypothesis 2b was not supported.

## 5. Discussion

As a productive deviant behavior, cyberloafing has been a widespread concern for enterprises because it represents a potential intervention point to improve productivity. The main purpose of the present study was to investigate the role of task crafting in reducing passive and active cyberloafing under a supportive context (supervisor developmental feedback). Based on the dual systems theory, the findings of our study show that task crafting is negatively related to passive cyberloafing and active cyberloafing. Supervisor developmental feedback moderates the relationship between task crafting and passive cyberloafing rather than active cyberloafing, and this relationship is stronger when supervisor developmental feedback is high rather than low.

### 5.1. Implications for Research

First, our study enriches the role of individual factors in minimizing cyberloafing [16]. Lim and Teo (2022) point out that few studies have explored how to reduce cyberloafing based on individual factors [6]. Our study introduced the theory of job crafting into cyberloafing and explored the interaction between task crafting and supervisor developmental feedback in passive and active cyberloafing to better understand the antecedents of these two types of cyberloafing. Our research suggests that employees can craft their tasks at work, and it has a significant negative effect on both passive and active cyberloafing. Task crafting negatively influences passive and active cyberloafing by reducing stress and satisfying intrinsic needs, respectively, although this study did not measure that process. However, the results are enough to suggest that the key to minimizing both types of cyberloafing lies in employee initiative. In other words, if employees are given the opportunity to shape their work, non-task-related behaviors may be reduced. Task crafting is a more beneficial intervention strategy for reducing cyberloafing than organizational control strategies, because well-designed tasks can not only reduce cyberloafing, but they can also have a positive impact on job performance. It lays a foundation for the intervention of cyberloafing in future management practice.

Second, the contribution of this study was to examine how the task crafting and cyberloafing association may vary under certain situational contexts by reflecting the person–situation interactive perspective and job crafting theory. We extended the job crafting theory to examine cyberloafing at work and how it varies depending on some important situational cues. High levels of supervisor developmental feedback play a critical role in strengthening the effect of employee task crafting, especially for passive cyberloafing, but not for active cyberloafing. The result reflects that the two forms of cyberloafing are differently sensitive to supportive environments. Specifically, minimizing active cyberloafing depends on individual effort, while minimizing passive cyberloafing requires individual effort and collaboration with the external environment factors. Supervisor developmental feedback can be translated into internal resources for employees, which can lead them to more effective allocation of attentional resources at work [45]. Indeed, the results are supported by Dierdorff and Jensen’s research (2018), in which a socially supportive context buffered the dysfunctional effects in job crafting and work productivity. However, the advancement made in the present study was to clarify that this social support can be concretized as supervisor developmental feedback [20].

The results show that supervisor developmental feedback does not buffer the relationship between task crafting and active cyberloafing, suggesting that the role of developmental feedback in moderating task crafting in relation to active cyberloafing may be more complex and difficult than that of passive cyberloafing. The realization of the task crafting lowering active cyberloafing may require employees with a high growth-need, and some internal needs can only be realized through the Internet. To be specific, first, supervisor developmental feedback only provides a potential opportunity to meet the intrinsic needs of employees, and whether they will eventually further meet the intrinsic needs through task crafting depends on the extent to which the intrinsic needs of employees are satisfied. In other words, if employees’ internal needs have been fully met through task crafting, supervisor developmental feedback will not push employees to further expand task boundaries. For high growth-need employees, who craft boundaries for themselves rather than responding to the task boundaries set for them [18], the decision on how to use supervisor developmental feedback lies with the “controlled processing” of the crafter, not the supervisor. Second, there are some intrinsic needs that cannot be replaced by work, such as the need for recovery (e.g., to reduce work fatigue), the need for socialization (e.g., to chat with friends), and the need for balancing work and family (e.g., to deal with urgent family matters via the Internet). This implies that necessary active cyberloafing at work may not be diminishing. This means that even with supervisor developmental feedback, task crafting does not reduce the necessary active cyberloafing.

Third, based on the dual systems theory, the exploration of the antecedents of passive and active cyberloafing expands the research in the field of cyberloafing. The concept of passive and active cyberloafing was proposed in 2018 [25], and its research tools were officially published in 2021 [37]. So far, only three articles have explored the antecedents or aftereffects of passive and active cyberloafing [14,21,22]. In conclusion, passive and active cyberloafing are common in real life, but the research evidence for these two types of cyberloafing is relatively scarce. This study is the first research to explore the organizational environments that can reduce passive and active cyberloafing.

Finally, our study provides a new cyberloafing scale to measure whether employees engage in cyberloafing intentionally or unintentionally. It is different from the previous cyberloafing scale, which focused on specific activities [3] and revealed the psychological basis for the occurrence of cyberloafing behavior. The distinction between passive and active cyberloafing can clarify whether cyberloafing behavior is based on automated processing or controlled processing. This scale complements the current scale in the field of cyberloafing to understand employees’ cyberloafing behavior from a new perspective. However, as described in this study, individuals’ motivations for engaging in active cyberloafing may be diverse, which are not addressed in this study and need to be further explored.

### 5.2. Implications for Practice

Current research provides some practical implications on how to reduce cyberloafing in the workplace, which can be costly for both organizations and individual employees. First, for organizations, cyberloafing interventions should consider the type of cyberloafing. In our study, passive and active cyberloafing interventions had different demands on the environment. Passive cyberloafing arises from an individual’s unconscious distraction at work. Therefore, not only does it require job crafting by the employees, but it also requires support from the external environment to reduce its own resource depletion and to help the employee become aware of or identify the non-work behavior. However, active cyberloafing only requires that organizations provide employees with sufficient creative opportunities and stimulate internal work motivation. Employees will automatically adjust their work to reduce their active withdrawal behaviors at work.

Second, because task crafting plays a critical role in reducing cyberloafing, organizations should increase employee engagement in task crafting. The organization can create a relaxed working atmosphere, providing employees with sufficient opportunities for crafting and job resources to support employees’ task crafting. In addition, short-term training can enhance employees’ participation in task crafting. For example, van Wingerden (2017) developed a four-week training based on the principle of positive goal setting that has been shown to increase employee engagement in task crafting and improve their job performance [46]. The training results not only showed significant effects 1 week after the intervention was completed, but also 1 year later [46].

Third, with intervention strategies for task crafting, particularly given the absence of supervisor if supervisors cannot provide developmental feedback, employees can receive support in other ways to facilitate task crafting, such as colleagues or other stakeholders [47]. Through vicarious learning, employees find out how to best adjust their work environment to their own abilities and preferences, facilitating person–environment fit and increasing the meaning of their work [48].

Finally, organizations need to select and develop suitable supervisors to provide feedback to employees. Since the quality of the feedback affects the effectiveness of the task crafting in reducing passive cyberloafing, it also places higher demands on the supervisor. Organizations need to screen candidates who are competent for this feedback task and provide supervisors with space and organizational support for sustainable growth.

### 5.3. Limitations and Future Research Directions

Our study has some limitations worth noting. First, in our study, the relationship between task crafting and two types of cyberloafing was preliminarily explored, but the different mediating mechanisms between task crafting and passive cyberloafing, and between task crafting and active cyberloafing, were not directly measured, which can be further discussed in future studies.

Second, the conclusions of this study are limited to employees who have a fixed workplace every day. Future studies could examine whether the conclusions of this study apply to other work patterns, such as autonomous work or remote work. In a less supervised environment, employees have more freedom. At this point, there are more opportunities for task crafting, but there are more opportunities for cyberloafing. In this case, it is unclear whether task crafting and supervisor developmental feedback can successfully reduce employees’ cyberloafing behavior.

Third, the effectiveness of task crafting in reducing cyberloafing may vary for different types of organizations, tasks, and employees, but was overlooked in our study because the job crafting theory states that task crafting can easily be carried out when employees perceive that they have enough job autonomy, such as low task interdependence and fewer department rules [18]. Take task interdependence as an example, because stronger task interdependence creates greater constraints and therefore less freedom to change task boundaries. In contrast, an employee whose task has little task interdependence with their colleagues (e.g., a hairdresser or a cleaner) has greater freedom to change the task boundaries. We hope that future research can specify the effects of task crafting on cyberloafing in different organizations, departments, and tasks, so as to facilitate the implementation of cyberloafing intervention strategies.

Finally, the motivations of employees and supervisors should also be taken into account in future studies. Employees engage in job crafting often based on different motivations, such as control over one’s work and building a positive self-image [18,49]. However, only high growth-need employees are able to undertake long-term task crafting [18]. Therefore, task crafting based on positive self-image may have a short-lived effect on reducing cyberloafing. In addition, the motivation of the supervisor to provide feedback also needs to be considered. Supervisor developmental feedback is usually future-oriented and employee-centered. However, the supervisor, as the coordinator of the organization and the employees, will inevitably provide advice from an organizational perspective. If supervisors do not support employees but instead make them feel stressed, feedback will have a negative effect. It may prevent employees from using feedback creatively in the crafting process. These questions are expected to be explored in future studies.

## 6. Conclusions

Overall, the results of the current study suggest that task crafting can be used as an intervention to reduce passive and active cyberloafing, and a supportive environment (supervisor developmental feedback) strengthens the impact of task crafting on passive cyberloafing. In addition, passive and active cyberloafing framework can be considered in cyberloafing research and intervention in the future.

## Figures and Tables

**Figure 1 behavsci-14-00960-f001:**
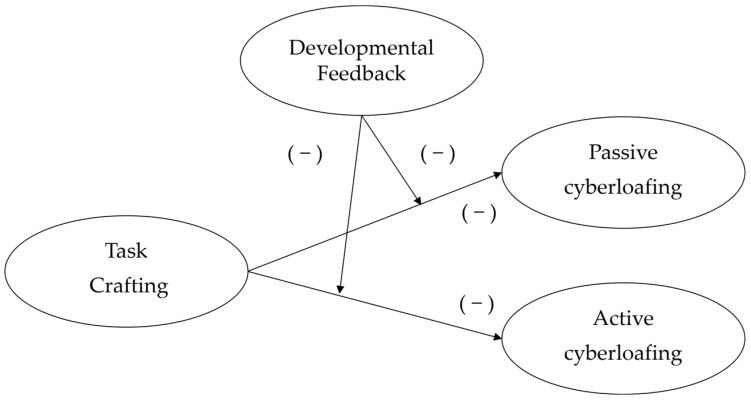
Hypothesized model of relationships among task crafting, developmental feedback, passive and active cyberloafing.

**Figure 2 behavsci-14-00960-f002:**
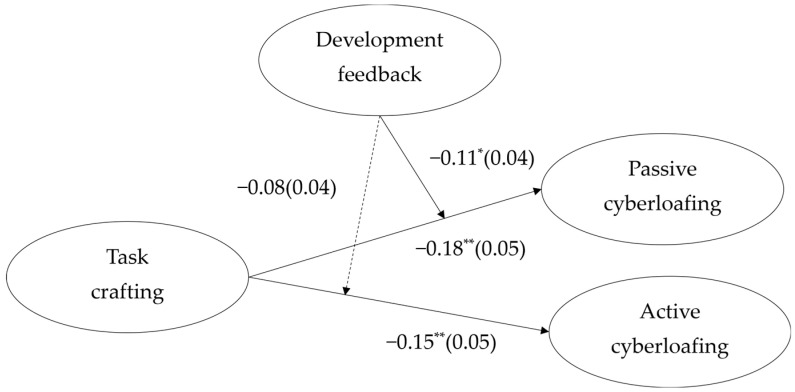
Unstandardized structural path coefficients and standard errors (in parentheses) for the structural equation model. Dashed arrows indicate non-significant effects. * *p* < 0.05, ** *p* < 0.01.

**Figure 3 behavsci-14-00960-f003:**
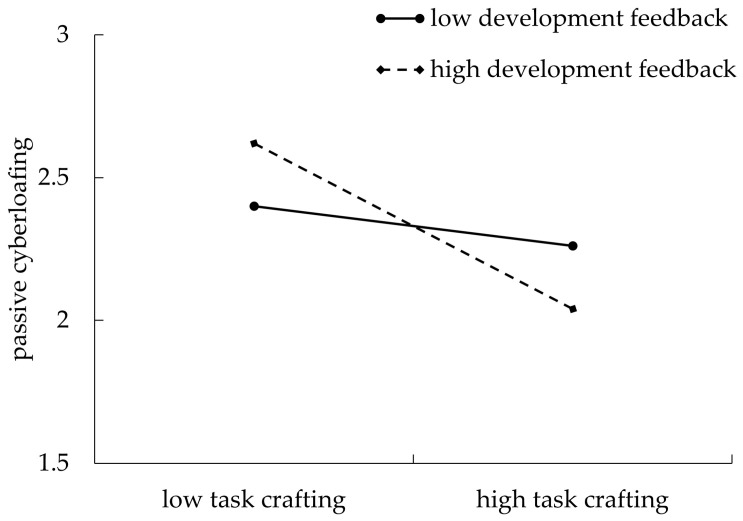
The moderating effect of supervisor development feedback on the relationship between task crafting and passive cyberloafing.

**Table 1 behavsci-14-00960-t001:** Descriptive statistics and correlation of passive–active cyberloafing (N = 224).

	M	SD	1	2	3	4	5	6	7	8	9	10
1 Cyberloafing	3.12	1.08										
2 Passive cyberloafing	3.09	1.20	0.93 **									
3 Active cyberloafing	3.14	1.14	0.92 **	0.70 **								
4 Lim’s cyberloafing	2.44	0.88	0.75 **	0.62 **	0.77 **							
5 Goal clarity	3.99	0.77	−0.11	−0.16 *	−0.04	0.04						
6 Job performance	3.49	0.73	−0.17	−0.20 **	−0.12	−0.03	0.57 **					
7 Gender	1.59	0.49	0.12	0.12	0.09	0.09	−0.05	−0.06				
8 Age	25.81	5.72	−0.11	−0.15 *	−0.05	−0.02	0.18 **	0.24 **	0.01			
9 Education level	3.62	0.71	0.06	0.08	0.02	0.11	0.06	0.06	0.10	0.18 **		
10 Rank	1.50	0.76	−0.14 *	−0.16 *	−0.10	0.02	0.25 **	0.33 **	−0.14 *	0.32 **	0.08	
11 Tenure	2.98	3.20	−0.07	−0.11	−0.02	0.04	0.19 **	0.27 **	−0.05	0.70 **	0.08	0.37 **

Note. * *p* ≤ 0.05, ** *p* ≤ 0.01.

**Table 2 behavsci-14-00960-t002:** Means, standard deviations, and correlations among variables.

Variable	*M*	*SD*	1	2	3	4
1. Task crafting	4.34	0.94				
2. Developmental feedback	3.15	1.03	0.27 **			
3. Passive cyberloafing	2.89	0.96	−0.14 **	−0.04		
4. Active cyberloafing	2.70	0.97	−0.12 **	−0.06	0.67 **	
5. Cyberloafing	2.80	0.88	−0.14 **	−0.05	0.91 **	0.92 **

Note. ** *p* ≤ 0.01.

## Data Availability

The data in this study are available from the corresponding authors upon reasonable request.

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
