# Peer review of "Start Task Crafting, Stay Away from Cyberloafing: The Moderating Role of Supervisor Developmental Feedback"

_behavsci, 2024, doi:10.3390/bs14100960_

Round 1
Reviewer 1 Report
Comments and Suggestions for Authors
Well-done.
Some suggestions to consider:
(1) Given the nature of work being either autonomous or remote, the work day is not confined into set hours like 8:30 - 5:00. As such, employees will "self-compensate", which means that they recognize periods of cyberloafing, but will make up the time and work outside of their typical work hours.
(2) Furthermore, given that there are increased work efficiencies that people use, employees might feel that they can work 6 hours a day (i.e., 8 hour work day minus 2 hours of intermittent cyberloafing) and get just as much accomplished as they could over a period of 8 hours, which they put in 10-20 years ago. For example, they can use ChatGPT to help draft or edit a report and use the time saved to engage in active or passive cyberloafing.
(3) I am wondering if employees who engage in active cyberloafing might feel differentially guilty from when they engage in passive cyberloafing. The former has greater intentionality.
In 4.3, there is a sentence "The results show that the model fits good". That should either be "the model fits well" or "the model fit is good"
Reviewer 2 Report
Comments and Suggestions for Authors
Although the study is situated within the framework of research on job crafting and dual-process theory to study some determinants of cyberloafing, the methodology chosen and the results achieved are a modest contribution to the production of knowledge about the phenomenon, which is reflected in the conclusions presented, which are short and linear, limiting themselves to suggesting that “task crafting can be used as an intervention to reduce passive and active cyberloafing”, without specifying in which type of organizations, in which sectors of activity or in which type of tasks.
Since it uses an (new!) assessment instrument, the Active and Passive Cyberloafing Scale (APCS), it would have been enriching to include in this article some validity studies on the instrument, just as it could have been a good contribution to also include a characterization of the sample with presentation of data relating, for example, to the variation of cyberloafing values according to criterion variables related to the sample (gender, age, activity sector, seniority, size of the organization, type of tasks, functions performed, etc.).
Without these developments, I do not believe that publication is justified.
As for formal aspects, I suggest the following corrections (The lines are not numbered om the paper):
1) Page 3 (lines 15, 17, 28), Page 6 (line 22), Page 9 (lines 31, 32), it is necessary to insert a space between the last word and the parenthesis where the author's name or the date is inserted. Please correct.
2) Pages 4, 5, 9 (on several lines), remove the comma placed between the author's name and the expression "et al."). Please correct.
3) Explain the acronyms used in lines 7 and 8 of page 7 (AIC and BIC). Please correct.
4) Use capital letters for each word in the name of the journals in reference 3. Please correct; 5) Eliminate excess capital letters in the titles of articles in references 4, 5, 14, 37, 42. Please correct.
Reviewer 3 Report
Comments and Suggestions for Authors
In the abstract, the authors should explicitly state the types of analysis conducted, the methods of data collection, and the research design used in the study.
In the introduction, they reference a survey: “A survey shows 67 percent of employees use the company’s Internet for non-work-related purpose, ranging from one hour to ten hours a week (Salary.com, 2014)”. Additional details should be provided by clarify the scope, participants, and conditions of the survey.
The authors stated in the introduction: “Third, based on the dual process theory, our study is the first to explore whether the factors that minimize passive and active cyberloafing are the same (Chen et al., 2022; Hai & Li, 2024)”. However, the factors under investigation are not mentioned.
The dual process theory and the cognitive processes related to cyberloafing should be described in greater depth, and the connection to this research should be made explicit. Additionally, the concept of “the need of recovery” should be incorporated into a theoretical framework.
Regarding the authors’ statement: “The model index of SEM model containing the interaction terms (AIC = 27130.23, BIC = 27404.26) is better than the baseline SEM model (AIC = 27134.45, BIC = 27399.65). The decrease of AIC and BIC indicates that the model has improved”, the authors suggest that the reduction in AIC and BIC indicates a model improvement. However, this result may be influenced by unobserved heterogeneity and missing covariate. The authors should provide a more thorough justification for these results, offering additional information and analysis. I think the author should at least include control variables. Ideally, if possible, it would be to consider workers and supervisor motivation in the model.
In practical terms, the authors should discuss strategies for enhancing employe engagement in task crafting, particularly given the absence of supervisor moderation effect.
In the discussion, the authors introduce the concept of value. The role of value is unclear. The authors provide little explanation regarding the statement that developmental feedback fails to moderate the relationship between task crafting and active cyberloafing. Within the theoretical framework, reference should be made to the role of values to justify the consideration mentioned in the discussion.
The limitations section should be expanded to address the concerns raised, providing more detail on potential shortcomings.
Round 2
Reviewer 2 Report
Comments and Suggestions for Authors
The review addressed the existing gaps.
The text added at the end (supplement) could be integrated into the article, however, if the editorial policy agrees, it corresponds to a practice of some journals, so it may be accepted.
Reviewer 3 Report
Comments and Suggestions for Authors
The authors have done a good job.
Author Response
Thank you for your help in improving the quality of our manuscripts. Thank you very much!